# The Kernel Density Integral Transformation

**Calvin McCarter**                                                  *mccarter.calvin@gmail.com*
*Boston, MA*

**Reviewed on OpenReview:** *https://openreview.net/forum?id=6OEcDKZj5j*

## Abstract

Feature preprocessing continues to play a critical role when applying machine learning and statistical methods to tabular data. In this paper, we propose the use of the kernel density integral transformation as a feature preprocessing step. Our approach subsumes the two leading feature preprocessing methods as limiting cases: linear min-max scaling and quantile transformation. We demonstrate that, without hyperparameter tuning, the kernel density integral transformation can be used as a simple drop-in replacement for either method, offering protection from the weaknesses of each. Alternatively, with tuning of a single continuous hyperparameter, we frequently outperform both of these methods. Finally, we show that the kernel density transformation can be profitably applied to statistical data analysis, particularly in correlation analysis and univariate clustering.

## 1 Introduction

Feature preprocessing is a ubiquitous workhorse in applied machine learning and statistics, particularly for structured (tabular) data. Two of the most common preprocessing methods are min-max scaling, which linearly rescales each feature to have the range $[0, 1]$, and quantile transformation (Bartlett, 1947; Van der Waerden, 1952), which nonlinearly maps each feature to its quantiles, also lying in the range $[0, 1]$. Min-max scaling preserves the shape of each feature's distribution, but is not robust to the effect of outliers, such that output features' variances are not identical. (Other linear scaling methods, such as $z$-score standardization, can guarantee output uniform variances at the cost of non-identical output ranges.) On the other hand, quantile transformation reduces the effect of outliers, and guarantees identical output variances (and indeed, all moments) as well as ranges; however, all information about the shape of feature distributions is lost.

In this paper, we observe that, by computing definite integrals over the kernel density estimator (KDE) (Rosenblatt, 1956; Parzen, 1962; Silverman, 1986) and tuning the kernel bandwidth, we may construct a tunable "happy medium" between min-max scaling and quantile transformation. Our generalization of these transformations is nevertheless both conceptually simple and computationally efficient, even for large sample sizes. On a wide variety of tabular datasets, we demonstrate that the kernel density integral transformation is broadly applicable for tasks in machine learning and statistics.

We hasten to point out that kernel density estimators of quantiles have been previously proposed and extensively analyzed (Yamato, 1973; Azzalini, 1981; Sheather, 1990; Kulczycki & DaWidowicz, 1999). However, previous works used the KDE to yield quantile estimators with superior statistical qualities. Statistical consistency and efficiency are desired for such estimators, and so the kernel bandwidth $h$ is chosen such that $h \to 0$ as sample size $N \to \infty$. However, in our case, we are not interested in estimating the true quantiles or the cumulative distribution function (c.d.f.), but instead use the KDE merely to construct a preprocessing transformation for downstream prediction and data analysis tasks. In fact, as we will see later, we choose $h$ to be large and non-vanishing, so that our preprocessed features deviate substantially from the empirical quantiles.

Our contributions are as follows. First, we propose the kernel density integral transformation as a feature preprocessing method that includes the min-max and quantile transforms as limiting cases. Second, we provide a computationally-efficient approximation algorithm that scales to large sample sizes. Third, we demonstrate the use of kernel density integrals in correlation analysis, enabling a useful compromise between

Pearson's $r$ and Spearman's $\rho$. Third, we propose a discretization method for univariate data, based on computing local minima in the kernel density estimator of the kernel density integrals.

## 2 Methods

### 2.1 Preliminaries

Our approach is inspired by the behavior of min-max scaling and quantile transformation, which we briefly describe below.

The min-max transformation can be derived by considering a random variable $X$ defined over some known range $[U, V]$. In order to transform this variable onto the range $[0, 1]$, one may define the mapping $S : \mathbb{R} \to [0, 1]$ defined as $x \to S(x) := \frac{x-U}{V-U}$, with the upper and lower bounds achieved at $x = U$ and $x = V$, respectively. In practice, one typically observes $N$ random samples $X_1, \ldots, X_N$, which may be sorted into order statistics $X_{(1)} \le X_{(2)} \le \cdots \le X_{(N)}$. Substituting the minimum and maximum for $U$ and $V$ respectively, we obtain the min-max scaling function

$$\hat{S}_N(x) := \begin{cases} \frac{x - X_{(1)}}{X_{(N)} - X_{(1)}}, & X_{(1)} \le x \le X_{(N)} \\ 0, & x \le X_{(1)} \\ 1, & X_{(N)} \le x. \end{cases} \tag{1}$$

Quantile transformation can be derived by considering a random variable $X$ with known continuous and strictly monotonically increasing c.d.f. $F_X : \mathbb{R} \to [0, 1]$. The quantile transformation (to be distinguished from the quantile function, or inverse c.d.f.) is identical to the c.d.f, simply mapping each input value $x$ to $F_X(x) := P[X \le x]$. One typically observes $N$ random samples $X_1, \ldots, X_N$ and obtains an empirical c.d.f. as follows:

$$\hat{F}_N(x) = \frac{1}{N} \sum_{n=1}^{N} I\{X_n \le x\} := \hat{P}[X \le x]. \tag{2}$$

The quantile transformation also requires ensuring sensible behavior despite ties in the observed data.

### 2.2 The kernel density integral transformation

Our proposal is inspired by the observation that, just as the quantile transformation is defined by a data-derived c.d.f., the min-max transformation can also be interpreted as the c.d.f. of the uniform distribution over $[X_{(1)}, X_{(N)}]$. We thus propose to interpolate between these via the kernel density estimator (KDE) (Silverman, 1986). Recall the Gaussian KDE with the following density:

$$\widehat{f}_h(x) = \frac{1}{N} \sum_{n=1}^{N} K_h(x - X_n) = \frac{1}{Nh} \sum_{n=1}^{N} K\left(\frac{x - X_n}{h}\right), \tag{3}$$

where Gaussian kernel $K(z) = \exp(-z^2/2)/\sqrt{2\pi}$ and $h > 0$ is the kernel bandwidth. Consider further the definite integral over the KDE above as a function of its endpoints:

$$P_h(a, b) := \int_a^b \widehat{f}_h(x) dx. \tag{4}$$

If we replace the empirical c.d.f. in Eq. (2) with the KDE c.d.f. using Eq. (4), we obtain the following:

$$\hat{F}_N^{\text{KDI,naive}}(x; h) := P_h(-\infty, x). \tag{5}$$

However, we change the integration bounds from $(-\infty, \infty)$ to $(X_{(1)}, X_{(N)})$, such that we obtain $\hat{F}_N^{\text{KDI}}(x) = 0$ for $x \le X_{(1)}$ and $\hat{F}_N^{\text{KDI}}(x) = 1$ for $X_{(N)} \le x$. To do this, while keeping $\hat{F}_N^{\text{KDI}}(x)$ continuous, we define the

final version of the kernel density integral (KD-integral) transformation as follows:

$$\hat{F}_N^{\text{KDI}}(x; h) := \begin{cases} \frac{P_h(X_{(1)}, x)}{P_h(X_{(1)}, X_{(N)})}, & X_{(1)} \le x < X_{(N)} \\ 0, & x < X_{(1)} \\ 1, & X_{(N)} \le x. \end{cases} \tag{6}$$

It can be seen that our proposed estimator matches the behavior of the min-max transformation and the quantile transformation at the extrema $X_{(1)}$ and $X_{(N)}$. Also, in practice, we parameterize $h = \alpha \hat{\sigma}_X$, where $\alpha$ is the *bandwidth factor*, and $\hat{\sigma}_X$ is an estimate of the standard deviation of $X$.

As the kernel bandwidth $h \to \infty$, the KD-integral transformation will converge towards the min-max transformation. Meanwhile, when $h \to 0$, the KD-integral transformation converges towards the quantile transformation. (Proofs are given in the Appendix.) Furthermore, as noted previously, $\hat{F}_N^{\text{KDQ,naive}}(x)$ is exactly the formula for computing the kernel density estimator of quantiles. However, in this paper, we propose (for the first time, to our knowledge) choosing a large kernel bandwidth. Rather than estimating quantiles with improved statistical efficiency as in (Sheather, 1990; Kulczycki & DaWidowicz, 1999), our aim is carry out a transformation that is an optimal compromise between the min-max transformation and the quantile transformation, for a given downstream task. As we will see in the experiment section, this optimal compromise is often struck at large bandwidths such as $h = 1 \cdot \hat{\sigma}_X$, even as the sample size $N \to \infty$.

## 2.3 Efficient computation

For supervised learning preprocessing, the KD-integral transformation must accommodate separate train and test sets. Thus, it is desirable that it be estimated quickly on a training set, efficiently (yet approximately) represented with low space complexity, and then applied quickly to test samples. For the Gaussian kernel, we compute and store KD-integrals at $R = \min(R_{\max} = 1000, N)$ reference points, equally spaced in $[0, 1]$. New points are transformed by linearly interpolating these KD-integrals. This approach has runtime complexity of $O(RN_{\text{train}})$ at training time and $O(\log(R)N_{\text{test}})$ at test time, and space complexity of $O(R)$. While this approach offers tunable control of the error and efficiency at test-time, the tradeoff between precision and training time is not ideal, as shown in Figure 1.

To address this, we applied the $\text{poly}(|x|)e^{-|x|}$ kernel (Hofmeyr, 2019) to our setting. The polynomial-exponential family of kernels, parameterized by the order $\kappa$ of the polynomial, can be

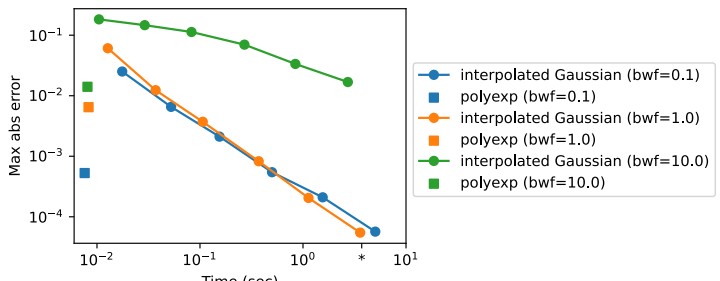

Figure 1: Precision versus runtime tradeoff for different bandwidth factors $\alpha \in \{0.1, 1, 10\}$. The data consisted of $N_{\text{train}} = 10{,}000$ training points sampled from LogNormal$(0, 1)$ and $N_{\text{test}} = 10{,}000$ equally-spaced test points within the range of the training data. The error is computed as the maximum absolute error between estimated values and those from the exact Gaussian KDE cdf. We compare against naively approximating the Gaussian KDE via sampling $S \in \{30, 100, 300, 10^3, 3 \cdot 10^3, 10^4\}$ without replacement to form the KDE, followed by interpolating with $R = 1000$ references as usual. The runtime using exact Gaussian cdf computation is depicted on the x-axis ("*"). Runtime was measured on a machine with a 2.8 GHz Core i5 processor.

evaluated via dynamic programming with runtime complexity $O(\kappa N)$, eliminating the quadratic dependence on the number of samples. We use order $\kappa = 4$, which yields a smooth kernel with infinite support that closely approximates the Gaussian kernel with an appropriate rescaling of the bandwidth (Hofmeyr, 2019). More details are given in the Appendix.

As shown in Figure 1, for 10k samples, this results in almost a 1000x speedup compared to exact evaluation of the Gaussian KDE cdf, across a range of bandwidth factors. It also offers a 10x to 100x training time

speedup at a fixed precision level, compared to naively speeding up the Gaussian KDE by subsampling data points then interpolating as usual.

Our software package, implemented in Python/Numba with a Scikit-learn (Pedregosa et al., 2011) compatible API, is available at https://github.com/calvinmccarter/kditransform.

### 2.4 Application to correlation analysis

In correlation analysis, whereas Pearson's $r$ (Pearson, 1895) is appropriate for measuring the strength of linear relationships, Spearman's rank-correlation coefficient $\rho$ (Spearman, 1904) is useful for measuring the strength of non-linear yet monotonic relationships. Spearman's $\rho$ may be computed by applying Pearson's correlation coefficient after first transforming the original data to quantiles or ranks $R(x) := N\hat{F}_N(x)$. Thus, it is straightforward to extend Spearman's $\rho$ by computing the correlation coefficient between two variables by computing their respective KD-integrals, then applying Pearson's formula as before. Like Spearman's $\rho$, it is apparent that ours is a particular case of a general correlation coefficient $\Gamma$ Kendall (1948). For $N$ samples of random variables $X$ and $Y$, the general correlation coefficient $\Gamma$ may be written as

$$\Gamma = \frac{\sum_{i,j=1}^{N} a_{ij} b_{ij}}{\sqrt{\sum_{i,j=1}^{N} a_{ij}^2 \sum_{i,j=1}^{N} b_{ij}^2}},$$

for $a_{ij} := r_j - r_i$, $b_{ij} := s_j - s_i$, where now $r_i$ and $s_i$ correspond to the KD-integrals of $X_i$ and $Y_i$, respectively.

Because ranks are robust to the effect of outliers, Spearman's $\rho$ is also useful as a robust measure of correlation; our proposed approach inherits this benefit, as will be shown in the experiments.

### 2.5 Application to univariate clustering

Here we apply KD-integral transformation to the problem of univariate clustering (a.k.a. discretization). Our approach relies on the intuition that local minima and local maxima of the KDE will tend to correspond to cluster boundaries and cluster centroids, respectively. However, naive application of this idea would perform poorly because low-density regions will tend to have many isolated extrema, causing us to partition low-density regions into many separate clusters. When we apply the KD-integral transformation, we draw such points in low-density regions closer together, because the definite integrals between such points will tend to be small. Then, when we form a KDE on these transformed points and identify local extrema, we avoid partitioning such low-density regions into many separate clusters.

Our proposed approach thus comprises three steps:

1. Compute $T_n = \hat{F}_N^{\text{KDQ}}(X_n)$, $\forall n \in \{1, \ldots, N\}$.

2. Form the kernel density estimator $\hat{f}_h(t)$ for $T_1, \ldots, T_N$. For this second KDE, select a vanishing bandwidth via Scott's Rule ($h = N^{-0.2} \sigma_X$) (Scott, 1992).

3. Identify the cluster boundaries from the local minima in $\hat{f}_h(t)$, and inverse-KD-integral-transform the boundaries for $T_n$ to obtain boundaries for clustering $X_n$.

## 3 Experiments

In this section, we evaluate our approach on supervised classification problems with real-world tabular datasets, on correlation analyses using simulated and real data, and on clustering of simulated univariate datasets with known ground-truth. We use the $\text{poly}(|x|)e^{-|x|}$ kernel with polynomial order $\kappa = 4$ in our experiments.

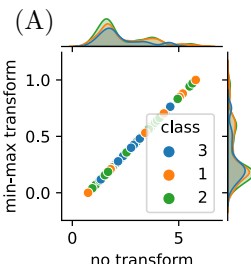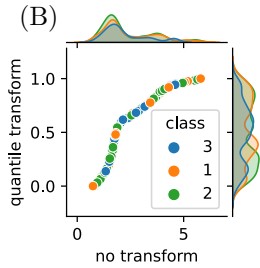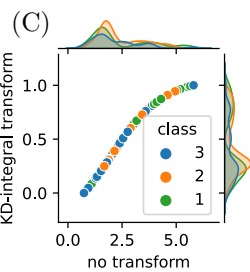

Figure 2: Comparison of (A) min-max scaling, (B) quantile transformation, and (C) KD-integral transformation on the `MalicAcid` feature in the Wine dataset. The horizontal density plots depict the distribution of the original data, while the vertical density plots show the distribution after each preprocessing step.

## 3.1 Feature preprocessing for supervised learning

### 3.1.1 Classification with PCA and Gaussian Naive Bayes

We first replicate the experimental setup of (Raschka, 2014), analyzing the effect of feature preprocessing methods on a simple Naive Bayes classifier for the Wine dataset (Forina et al., 1988). In addition to min-max scaling, used in (Raschka, 2014), we also try quantile transformation and our proposed KD-integral transformation approach. In Figure 2, we illustrate the effect of min-max scaling, quantile transformation, and KD-integral transformation on the `MalicAcid` feature on this dataset. We see that KD-integral transform in Figure 2(C) is concave for inputs $> 3$, compressing outliers together, while preserving the bimodal shape of the feature distribution. It is substantially smoother than the empirical quantile transform in Figure 2(B).

We examine the accuracy resulting from each of the preprocessing methods, followed by (as in (Raschka, 2014)) principle component analysis (PCA) with 2 components, followed by a Gaussian Naive Bayes classifier, evaluated via a 70-30 train-test split. For the KD-integral transformation, we show results for the default bandwidth factor of 1, for a bandwidth factor chosen via inner 30-fold cross-validation, and for a sweep of bandwidth factors between 0.1 and 10. The accuracy, averaged over 100 simulated train-test splits, is shown in Figure 3(A). We repeat the above experimental setup for 3 more popular tabular classification datasets: Iris (Fisher, 1936), Penguins (Gorman et al., 2014), Hawks (Cannon et al., 2019), shown in Figure 3(B-D), respectively.

Compared to min-max scaling and quantile transformation, our tuning-free proposal wins on Wine; on Iris, it is sandwiched between min-max scaling and quantile transformation; on Penguins, it wins over both min-max and quantile transformation; on Hawks, it ties with min-max while quantile transformation struggles. We also compare to $z$-score scaling; our tuning-free method beats it on Wine and Iris, loses to it on Penguins, and ties with it on Hawks. Overall, among the tuning-free approaches, ours comes in first, second, second, and second, respectively. Our approach with tuning is always non-inferior or better than min-max scaling and quantile transformation, but loses to $z$-score scaling on Penguins; however, note that $z$-score scaling performs poorly on Wine and Iris.

### 3.1.2 Supervised regression with linear regression

We compare the different methods on two standard regression problems, California Housing (Pace & Barry, 1997) and Abalone (Nash et al., 1995), with results depicted in Figure 4. Here we measure the root-mean-squared error on linear regression after preprocessing, with cross-validation setup as before. KD-integral, both with default bandwidth and with cross-validated bandwidth, outperformed the other approaches. Even in CA Housing, where quantile transformation outperforms min-max scaling due to skewed feature distributions, there is evidently benefit to maintaining a large enough bandwidth to preserve the distribution's shape. In both regression datasets, the performance improvement offered by KD-integral exceeds the difference in performance between linear and quantile transformations. We perform additional experiments on CA

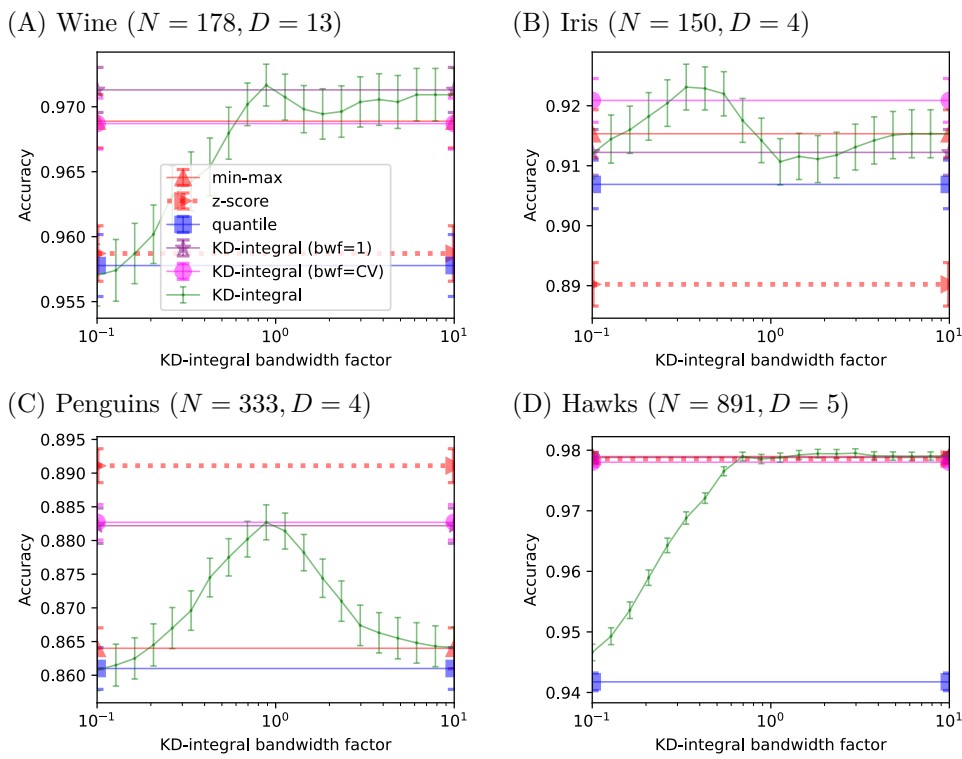

Figure 3: Accuracy on supervised classification problems for different feature preprocessing methods. Results are shown for Wine (A), Iris (B), Penguins (C), and Hawks (D); higher accuracy is better. Accuracy is shown as a horizontal line for min-max scaling, $z$-score scaling, quantile transformation, KD-integral with the default bandwidth factor $\alpha = 1$, and KD-integral with bandwidth selected via CV. In green, we show accuracy as a function of KD-integral bandwidth factor $\alpha$. Error bars depict the standard deviation over 100 simulations

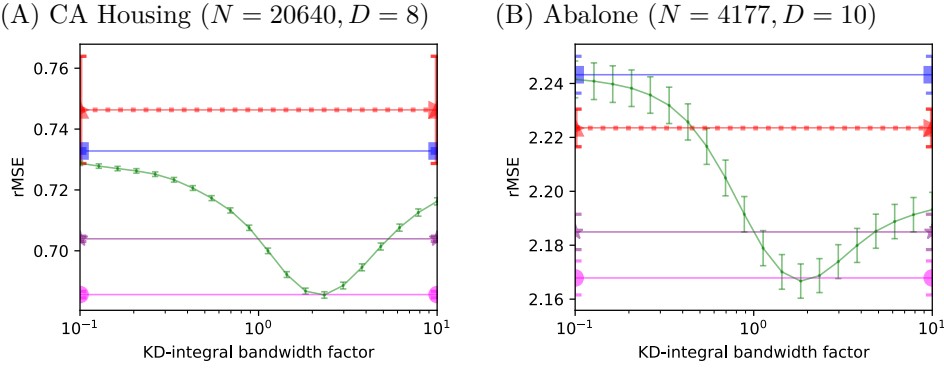

Figure 4: Root mean-squared error (rMSE) on supervised regression problems for different feature preprocessing methods. Results are shown for California housing (A), and Abalone (B); lower rMSE is better. Performance is shown as a horizontal line for min-max scaling, quantile transformation, KD-integral with the default bandwidth factor $\alpha = 1$, and KD-integral with bandwidth selected via CV. In green, we show rMSE as a function of KD-integral bandwidth factor $\alpha$.

Table 1: Performance of preprocessing methods on Small Data Benchmarks, as measured by area under the Receiver Operating Characteristic curve (ROC AUC). The columns display the mean and standard deviation of the ROC AUC, computed over 142 datasets in the benchmark. We also show $p$-values from one-sided Wilcoxon signed-rank test comparisons.

| METHOD | Mean (StdDev) ROC AUC | $p$ (vs KDI ($\alpha$=1)) | $p$ (vs KDI ($\alpha$=CV)) |
|---|---|---|---|
| Min-max | 0.864 (0.132) | $1.6 \times 10^{-3}$ | $5.1 \times 10^{-8}$ |
| Quantile | 0.866 (0.131) | $7.9 \times 10^{-3}$ | $2.9 \times 10^{-4}$ |
| KD-integral ($\alpha = 1$) | 0.868 (0.129) | | $3.7 \times 10^{-6}$ |
| KD-integral ($\alpha = CV$) | 0.869 (0.129) | | |

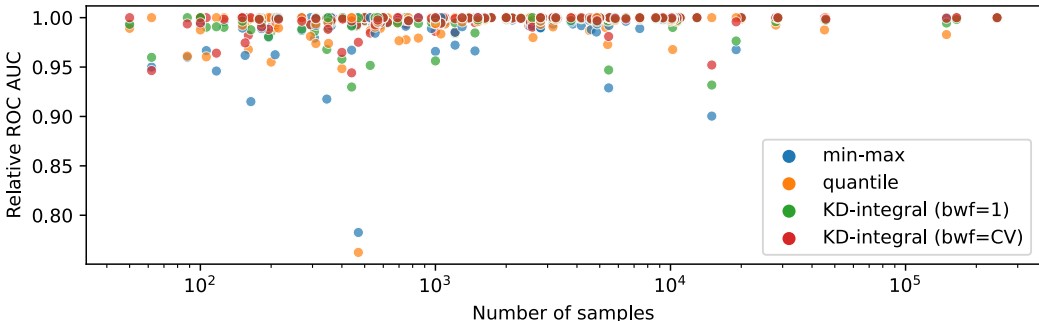

Figure 5: Performance of preprocessing methods on Small Data Benchmarks, plotted against dataset size. For each dataset, we compute the relative ROC AUC for a given method as its own ROC AUC, divided by the maximum ROC AUC over all preprocessing methods for that dataset.

Housing, with data subsampling, showing that the chosen bandwidth remains constant regardless of sample size; see the Appendix (A.3).

### 3.1.3 Linear classification on Small Data Benchmarks

We next compared preprocessing methods on a dataset-of-datasets benchmark, comprising 142 tabular datasets, each with at least 50 samples. We replicated the experimental setup of the Small Data Benchmarks (Feldman, 2021) on the UCI++ dataset repository (Paulo et al., 2015). In (Feldman, 2021), the leading linear classifier was a support vector classifier (SVC) with min-max preprocessing, to which we appended SVC with quantile transformation and SVC with KD-integral transformation. As in (Feldman, 2021), for each preprocessing method, we optimized the regularization hyperparameter $C \in \{10^{-4}, 10^{-3}, \ldots, 10^2\}$, evaluating each method via one-vs-rest-weighted ROC AUC, averaged over 4 stratified cross-validation folds. For the latter, we measured results with both the default bandwidth factor $\alpha = 1$ (so as to give equal hyperparameter optimization budgets to each approach), and with cross-validation grid-search to select $C$ and $\alpha \in \{3^{-1}, 3^0, 3^1\}$. Our results are summarized in Table 1. We see that KD-integral transformation provides statistically-significant greater average ROC AUC, with less variance, at the same tuning budget as the other approaches; further improvement is provided by tuning. We further analyzed the performance of the different methods in terms of the number of samples in each dataset in Figure 5. Plotting the relative ROC AUC against the number of samples $N$, we see that our proposed approach is particularly helpful in avoiding suboptimal performance for small-$N$ datasets.

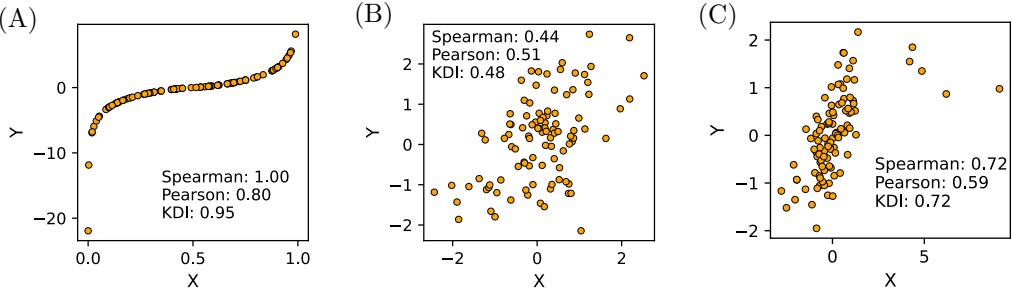

Figure 6: Illustration of correlation analysis using Pearson's $r$, Spearman's $\rho$, and our proposed approach, for simulated data. Three scenarios are depicted: (A) nonlinear yet monotonic relationship, (B) noisy linear relationship, and (C) linear relationship corrupted by outliers.

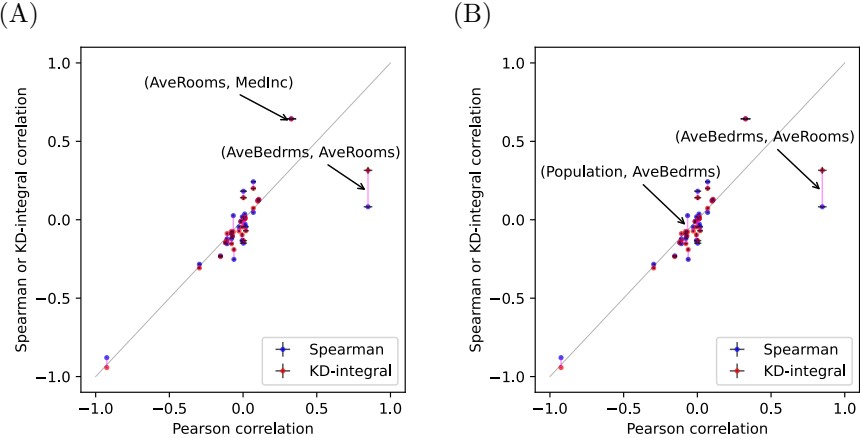

Figure 7: Correlation coefficients derived from the California housing dataset. The pink line depicts the gap between Spearman and KD-integral correlations, while the distance from the gray line shows how far each are from the Pearson correlation. Both (A) and (B) contain the same data, but top disagreements between ours and Pearson's, and ours and Spearman's are highlighted separately in (A) and (B), respectively. The error bars in black depict the standard deviation of each correlation, computed from 100 bootstrap simulations.

## 3.2 Correlation analysis

To provide a basic intuition, we first illustrate the different methods on synthetic datasets shown in Figure 6, replicating the example from (Wikipedia, 2009). When two variables are monotonically but not linearly related, as in Figure 6(A), the Spearman correlation exceeds the Pearson correlation. In this case, our approach behaves similarly to Spearman's. When two variables have noisy linear relationship, as in Figure 6(B), both the Pearson and Spearman have moderate correlation, and our approach interpolates between the two. When two variables have a linear relationship, yet are corrupted by outliers, the Pearson correlation is reduced due to the outliers, while the Spearman correlation is robust to this effect. In this case, our approach also behaves similarly to Spearman's.

Next, we perform correlation analysis on the California housing dataset, containing district-level features such as average prices and number of bedrooms from the 1990 Census. Overall, the computed correlation coefficients are typically close, with only a few exceptions, as shown in Figure 7. Our approach's top two disagreements with Pearson's are on (`AverageBedrooms`, `AverageRooms`) and (`MedianIncome`, `AverageRooms`). Our approach's top two disagreements with Spearman's are on (`AverageBedrooms`, `AverageRooms`) and (`Population`, `AverageBedrooms`). We further observe that KD-integral-based correlations typically, but not always, lie between the Pearson and Spearman correlation coefficients.

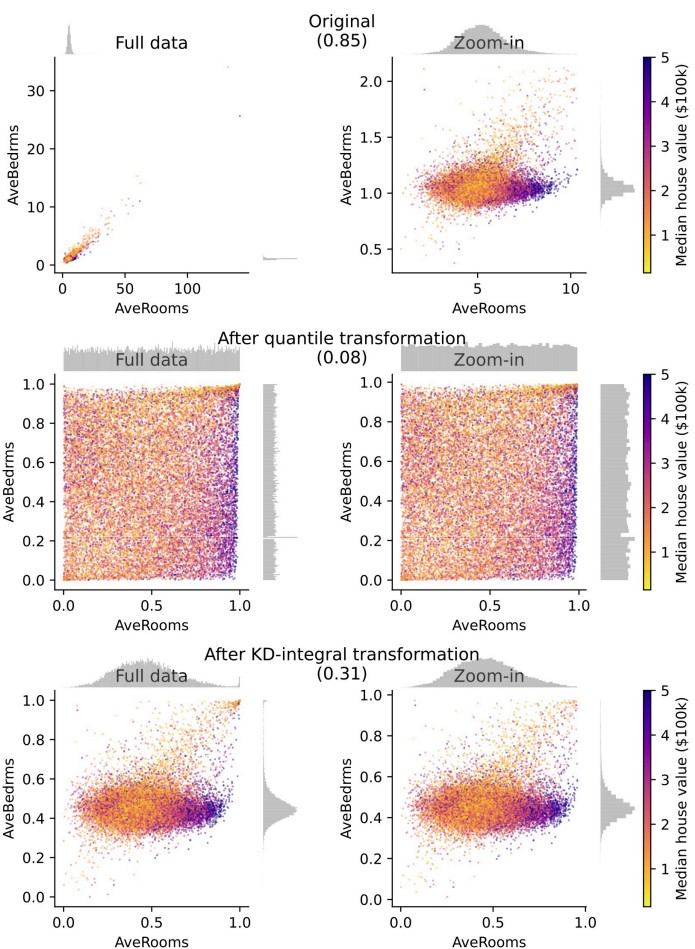

Figure 8: Correlation analysis for (`AverageBedrooms`, `AverageRooms`) in the California housing dataset. The rows, from top to bottom, correspond to original data, quantile transformation, and KD-integral transformation. The full dataset is shown on the left, while outliers are excluded on the right. Each district is colored by its median house value. In parenthesis above each row are the Pearson (0.85), Spearman (0.08), and KD-integral (0.31) estimated correlations between the variables.

We analyze the correlation disagreements for (`AverageBedrooms`, `AverageRooms`) in Figure 8. From the original data, it is apparent that `AverageBedrooms` and `AverageRooms` are correlated, whether we examine the full dataset or exclude outlier districts. This relationship was obscured by quantile transformation (and thus, by Spearman correlation analysis), whereas it is still noticeable after KD-integral transformation.

We repeat the analysis of disagreement for (`MedianIncome`, `AverageRooms`) and (`Population`, `AverageBedrooms`) in Figure 9. For the former disagreement, our approach agrees with quantile transformation-based analysis, identifying the typical positive dependence between median income and average rooms, by reducing the impact of districts with extremely high average rooms. For the latter disagreement, our approach agrees with original data-based analysis, identifying the negative relationship one would expect to observe between district population (and therefore density) and the average number of bedrooms.

## 3.3 Clustering univariate data

In this experiment, we generate five separate synthetic univariate datasets, each sampled according to the following mixture distributions:

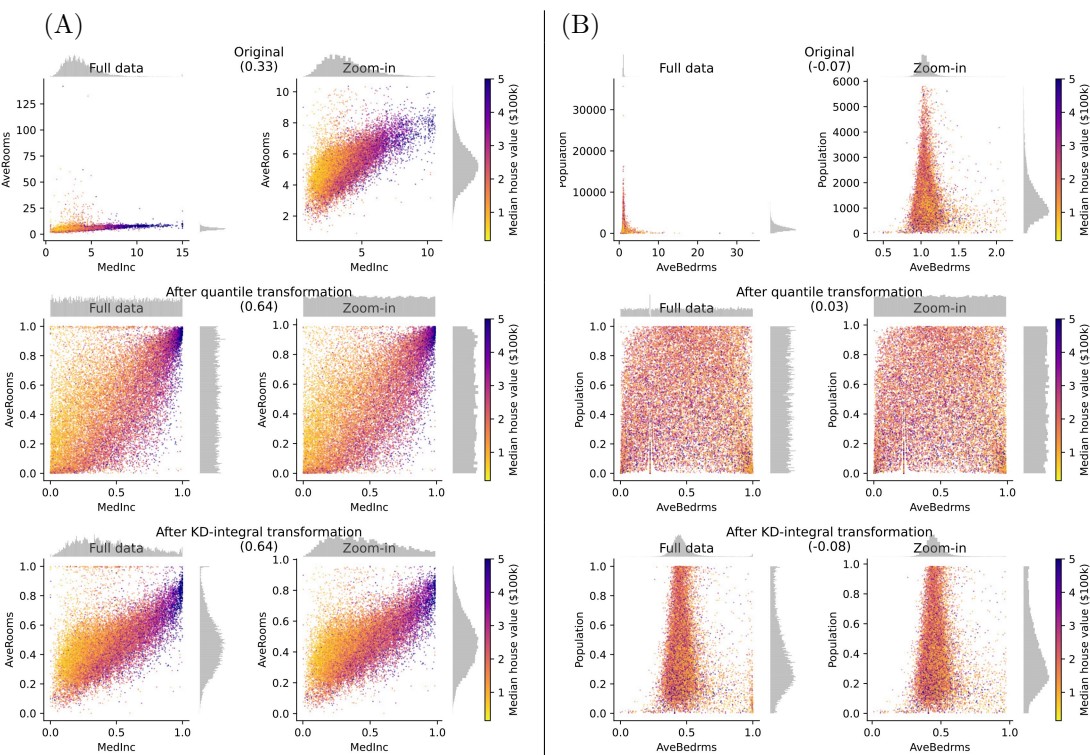

Figure 9: Correlation analysis for (`MedianIncome`, `AverageRooms`) (A) and (`Population`, `AverageBedrooms`) (B) in the California housing dataset. See Figure 8 for an explanation of the plot.

- $0.55 * \mathcal{N}(\mu = 1, \sigma = 0.75) + 0.30 * \mathcal{N}(\mu = 4, \sigma = 1) + 0.15 * \text{Unif}[a = 0, b = 20]$

- $0.45 * \mathcal{N}(\mu = 1, \sigma = 0.5) + 0.45 * \mathcal{N}(\mu = 4, \sigma = 1) + 0.10 * \text{Unif}[a = 0, b = 20]$

- $0.67 * \mathcal{N}(\mu = 1, \sigma = 0.5) + 0.33 * \mathcal{N}(\mu = 4, \sigma = 1)$

- $0.8 * \text{Exp}(\lambda = 1) + 0.2 * [10 + \text{Exp}(\lambda = 4)]$

- $0.5 * \text{Exp}(\lambda = 8) + 0.5 * [100 - \text{Exp}(\lambda = 5)].$

We compare our approach to five other clustering algorithms: KMeans with $K$ chosen to maximize the Silhouette Coefficient (Rousseeuw, 1987), GMM with $K$ chosen via the Bayesian information criterion (BIC), Bayesian Gaussian Mixture Model (GMM) with a Dirichlet Process prior, Mean Shift clustering (Comaniciu & Meer, 2002), and HDBSCAN (Campello et al., 2013; McInnes et al., 2017) (with `min_cluster_size=5, cluster_selection_epsilon=0.5`), and local minima of an adaptive-bandwidth KDE with adaptive `sensitivity=0.5` (Abramson, 1982; Wang & Wang, 2007).

In Figure 10, we compare the ground-truth cluster identities of the data with the estimated cluster identities from each of the methods, for $N = 500$ samples. On the top row, we depict the true clusters, as well as the true number of mixture components. On each of the following rows, we show the distributions of estimated clusters for each the methods. We see that our approach is the only method to correctly infer the true number of mixture components, and that it partitions the space similarly to ground-truth. GMM with BIC performed well on the first three datasets, but not on the last two. Meanwhile, KMeans performed well on the last three datasets but not on the first two. Bayesian GMM tended to slightly overestimate the number of components, while MeanShift and HDBSCAN (even after excluding the samples it classified as noise) tended to aggressively overestimate.

We include results for an ablation of our proposed approach, in which we define separate clusters at the local minima of the KDE of unpreprocessed inputs, rather than on the KDE of KD-integrals. We see that the

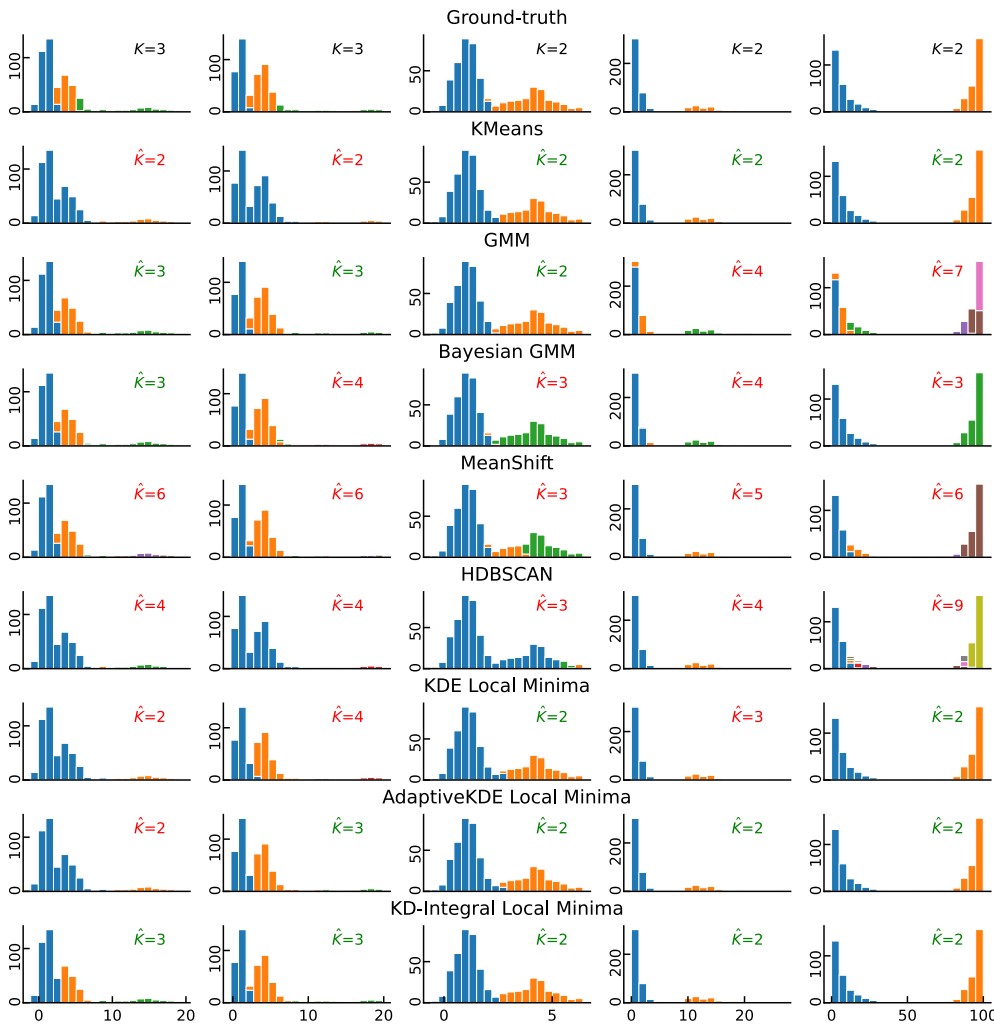

Figure 10: Clustering performance for five datasets (one dataset per column), generated with $N = 500$ samples. On each of the rows, for the different methods, we indicate the estimated number of components (green if correct, red if incorrect).

ablated method fails on the first, second, and fourth datasets, where there is a large imbalance between the mixture weights of the cluster components. Using the adaptive-bandwidth KDE fixes the second and fourth dataset but not the first.

We repeated the above experimental setup, this time varying the number of samples $N \in \{100, 200, 500, 1000, 2000, 5000\}$, and performing 20 independent simulations per each setting of $N$. For each simulation, we recorded whether the true $K$ and estimated $\hat{K}$ number of components matched, as well as the the adjusted Rand index (ARI) between the ground-truth and estimated cluster labelings. The results, averaged over 20 simulations, are shown in Figure 11. Our approach is the only method to attain high accuracy across all settings when $N > 1000$. Alternative methods behave inconsistently across datasets or across varying sample sizes. For the first two datasets with small $N$, KD-integral is outperformed by GMM and Bayesian GMM. However, GMM struggles on the first two datasets for large $N$, and on the fourth and fifth datasets; meanwhile, Bayesian GMM struggles on all datasets for large $N$. The only approach that comes close to KD-integral is using local minima of the adaptive KDE; however, it struggled with smaller sample sizes on the first and second datasets.

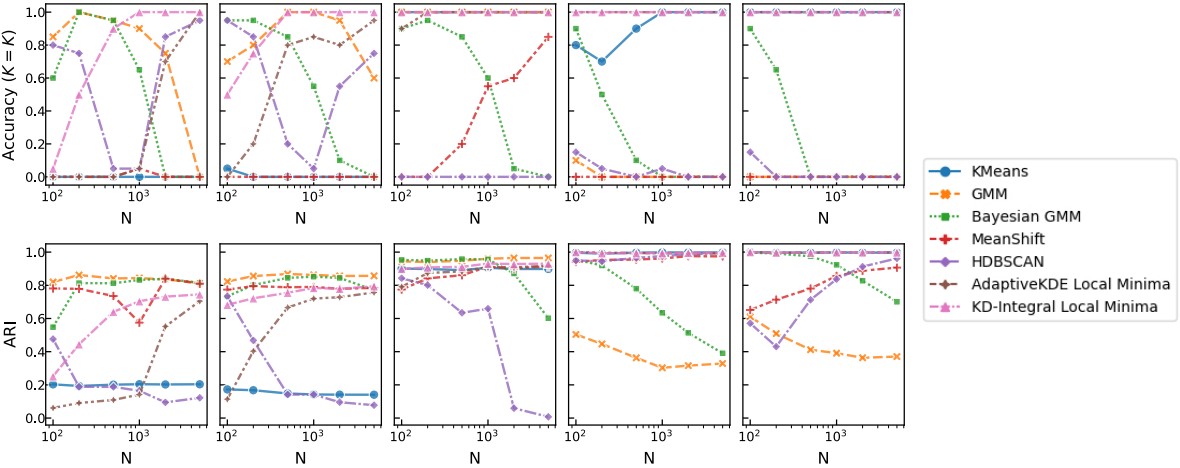

Figure 11: Performance at clustering univariate features, for varying number of samples $N$. The top row depicts the fraction of simulations in which the estimated number of mixture components equaled the true number. The bottom row depicts the average ARI between the ground-truth clustering and the estimated clusters. Each of the five columns corresponds to five mixture distributions described in the text and depicted on the top row of histograms.

## 4    Related Work

As far as we know, the use of kernel density integrals to interpolate between min-max scaling and quantile transformation has not been previously proposed in the literature. Similarly, we are not aware of kernel density integrals being proposed to balance between the strengths of Pearson's $r$ and Spearman's $\rho$.

Our approach is procedurally similar to copula transformations in statistics and finance (Cherubini et al., 2004; Patton, 2012). But because we have a different goal, namely a generic feature transformation that is to be extrinsically optimized and evaluated, our proposed approach has a markedly different effect. Besides the small adjustment from $\hat{F}^{\text{KDI,naive}}$ (which is procedurally identical to the kernel density copula transformation (Gourieroux et al., 2000)) to $\hat{F}^{\text{KDI}}$, our proposal aims at something quite different from copula transforms. The copula literature ultimately aims at transforming to a particular reference distribution (e.g., uniform or Gaussian), with the KDE used in place of the empirical distribution merely for statistical efficiency, thus choosing a consistency-yielding bandwidth (Gourieroux et al., 2000; Fermanian & Scaillet, 2003). We depart from this choice, finding that a large bandwidth that preserves the shape of the input distribution is frequently optimal for settings (e.g. classification) where marginal distributions need not have a given parametric form.

Our proposed approach for univariate clustering is similar in spirit to various density-based clustering methods, including mean-shift clustering (Comaniciu & Meer, 2002), level-set trees (Schmidberger & Frank, 2005; Wang & Huang, 2009; Kent et al., 2013), and HDBSCAN (Campello et al., 2013). However, such methods tend to leave isolated points as singletons, while joining points in high-density regions into larger clusters. To our knowledge, our approach for compressing together such isolated points has not been previously considered.

The kernel density estimator (KDE) was previously proposed (Flores et al., 2019) in the context of discretization-based preprocessing for supervised learning. However, their method did not use kernel density integrals as a preprocessing step, but instead employed a supervised approach that, for a multiclass classification problem with $C$ classes, constructed $C$ different KDEs for each feature.

## 5    Discussion

**Practical Recommendations**    We recommend that if one must employ a feature preprocessor without any tuning or comparisons between different preprocessing methods, then KD-integral with bandwidth-factor of 1 is the best one-shot option. If instead tuning is possible, we suggest comparing the performance of

KD-integral, with a log-space sweep of $\alpha \in [0.1, 10]$, in addition to the other popular preprocessing methods. Finally, we recommend that one always estimate our transformation with the order-4 polynomial-exponential kernel approximation of the Gaussian kernel, and represent it with $R_{\max} = 1000$ reference points.

**Limitations**  Our approach for supervised preprocessing is limited by the fact that, even though the bandwidth factor is a continuous parameter, its selection requires costly hyperparameter tuning. Meanwhile, our proposed approach would benefit from further theoretical analysis, as well as investigation into extensions for multivariate clustering.

**Future Work**  In this paper, we have focused on per-feature transformation. But, especially in genomics, it is common to perform per-sample quantile normalization (Bolstad et al., 2003; Amaratunga & Cabrera, 2001), in which features for a single sample are mapped to quantiles computed across all features; this would benefit from further study. Also, future work could study whether kernel density integrals could be profitably used in place of vanilla quantiles outside of classic tabular machine learning problems. For example, quantile regression (Koenker & Bassett Jr, 1978) has recently found increasing use in conformal prediction (Romano et al., 2019; Liu et al., 2022), uncertainty quantification (Jeon et al., 2016), and reinforcement learning (Rowland et al., 2023).

## 6  Conclusions

In this paper, we proposed the use of the kernel density integral transformation as a nonlinear preprocessing step, both for supervised learning settings and for statistical data analysis. In a variety of experiments on simulated and real datasets, we demonstrated that our proposed approach is straightforward to use, requiring simple (or no) tuning to offer improved performance compared to previous approaches.

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
