# OpenReview forum: "The Kernel Density Integral Transformation"
_TMLR — Accepted by TMLR_

### Review · Reviewer_qhJo · 2023-08-27

**Summary Of Contributions:**

The paper introduces a monotonic univariate transformation based on a min-max scaling of a kernel cdf evaluated at the sample. Traversing the values of the bandwidth from the infinitesimal to infinite, the transformation interpolates between quantile transformation and min-max scaling.

Some heuristic discussion relating to its potential advantages over these other two, as well as other common transformations, plus experiments documenting its potential usefulness in correlation analysis, univariate clustering and as a pre-processing tool for supervised learning are provided.

**Audience:**

Yes

**Claims And Evidence:**

No

**Requested Changes:**

Since the proposal is minimal in terms of a methodological and/or theoretical contribution, I believe it lives and dies on the persuasiveness of the experimental results. At the moment, although there are some nice observations coming through, the manner in which the experiments are conducted and the results displayed would benefit from considerably more work. I will note some possibilities below:

- Investigate the usefulness of tuning the bandwidth using, e.g., cross-validation to improve performance of supervised models using the porposed transformation. E.g. in the transformation + PCA + Naive Bayes. Note that I fully appreciate the need to automatic transformations which don't need tuning, at the moment the improvement here is extremely minimal and certainly not outside of what would reasonably be expected due to sampling variation. I am not suggesting you remove the automated (i.e., fixed scaling) results but as mentioned above the results need to be more persuasive to justify and entire publication.
- Use a log scale for Figure 3 since currently it is hard to see what is going on for small N
- Add some sort of resampling, e.g. bootstrap, to 3.2 to assess the sampling variation. In the pair of variables to which I directed a query earlier, population density and average bedrooms, the values are very close to zero and so could easily flip signs for different samples. At the moment, even if I am wrong about the intuitive sign of the relationship, the results are not persuasive due to the small magnitude of the correlations.
- The univariate clustering results are not persuasive. There are some methods designed for univariate clustering and none are included. It also seems unreasonable to simply use standard methods designed for primarily multivariate problems in a univariate setting without sensible modifications. Furthermore, DBSCAN is supposed to "find noise" and so saying that identifying very small collections of points which should have been merged into clusters is rather unfair.

**Strengths And Weaknesses:**

Strengths:
- The ubiquity of univariate scaling and transformations makes new alternatives to the common approaches of potential use.
- The intuitive connection to quantile and min-max scaling at the two extremes of the bandwidth makes the approach fairly accessible to non-experts.

Weaknesses:
- As a contribution the proposal is fairly minimal and lacks theoretical backing.
- The transformation seems quite similar to a kernel based copula transformation. Clearly since the transformation "maintains" some of the shape of the true distribution, where copulas aim for equivalent marginals, they are theoretically distinct. However copula transformations based on an over-smoothed kernel cdf will likely be very similar to the proposal. Ultimately the only difference between the proposal and a kernel based uniform copula is the fact that the proposal forces the transformation to span the interval [0, 1] which, while it may be beneficial, is a minimal contribution.
- The "speed up" using subsampling and a grid of reference points seems very suboptimal when there is considerable literature on efficient and highly accurate univariate kernel smoothing, e.g., [1,2,3].
- I'm not sure if this is just a typo, but it seems like a poor choice to set the bandwidth proportional to the VARIANCE of the sample, and not the standard deviation. Every piece of theory suggests proportionality to the scale of the variable and not squared scale. I appreciate the authors are distancing their approach from theory, but I presume they would want their method to be scale invariant at the very least.
- On the experimental side:
-- although the discussion about the tendency for "other" density based clustering methods to create small clusters in low density regions, it is very unlikely in my opinion that popular density based methods would be applied in the standard formulations to univariate data due to how much more simple the clustering problem is. Indeed something like a k-nn based density method or a simple univariate kde with adaptive bandwidth would similarly overcome this "problem".
-- The authors say that it is intuitively the case that population density and average number of rooms in houses are negatively correlated. I don't understand why this would not be opposite. Surely a higher density of people in an area requires more people per domicile, and hence more bedrooms on average? Please correct me if I am overlooking something obvious?
-- The experiments in general would benefit from some accommodation of sampling variation, using, e.g. bootstrap or some sort of stratified bootstrap when there are outliers present which should be retained.

Strength/Weakness:
- The additional "tuning parameter" being the bandwidth in the kernel cdf may be beneficial as an extra parameter to introduce complexity to simple (e.g., linear) models and potential enhance performance. At the same time there is invariably a cost of an additional hyperparameter, both in terms of model interpretability and computational demands associated with cross-validation or similar.

[1] Raykar, Vikas C., Ramani Duraiswami, and Linda H. Zhao. "Fast computation of kernel estimators." Journal of Computational and Graphical Statistics 19.1 (2010): 205-220.
[2] Hofmeyr, David P. "Fast exact evaluation of univariate kernel sums." IEEE transactions on pattern analysis and machine intelligence 43.2 (2019): 447-458.
[3] Fan, Jianqing, and James S. Marron. "Fast implementations of nonparametric curve estimators." Journal of computational and graphical statistics 3.1 (1994): 35-56.

---

> ### Author Response · Authors · 2023-09-18
> **Re: Review of Paper1458 by Reviewer qhJo**
>
> We thank you for your time and attention.
>
> > "The transformation seems quite similar to a kernel based copula transformation."
>
> We added to following to Related Work to clarify this matter: "Our approach is procedurally similar to copula transformations in statistics and finance (Cherubini et al., 2004; Patton, 2012). But because we have a different goal, namely a generic feature transformation that is to be extrinsically optimized and evaluated, our proposed approach has a markedly different effect. Besides the small adjustment from F^{KDI,naive} (which is procedurally identical to the kernel density copula transformation (Gourieroux et al., 2000)) to F^{KDI}, our proposal aims at something quite different from copula transforms.The copula literature ultimately aims at transforming to a particular reference distribution (e.g., uniform or Gaussian), with the KDE used in place of the empirical distribution merely for statistical efficiency, thuse choosing an AMISE-minimizing and consistency-yielding bandwidth (Gourieroux et al., 2000; Fermanian & Scaillet, 2003). We depart from this choice, finding that a large bandwidth that preserves the shape of the input distribution is frequently optimal for settings (e.g. classification) where marginal distributions need not have a given parametric form."
>
> > "The "speed up" using subsampling and a grid of reference points seems very suboptimal"
>
> Thank you for pointing this out. We have entirely redone that section, utilizing and analyzing the effectiveness of the approach in (Hofmeyr, 2019) which you shared.
>
>  > "I'm not sure if this is just a typo, but it seems like a poor choice to set the bandwidth proportional to the VARIANCE"
>
> Thank you. This typo has been corrected in the updated version.
>
> > "a simple univariate kde with adaptive bandwidth would similarly overcome this 'problem'" and "There are some methods designed for univariate clustering and none are included."
>
> We have added a baseline approach using local minima on the adaptive bandwidth KDE in the revised manuscript. We found that our proposed KD approach outperforms it for small sample sizes on datasets (1) and (2), while having the same performance elsewhere.
>
> > "Surely a higher density of people in an area requires more people per domicile, and hence more bedrooms on average?"
>
> In the USA, single-family homes tend to have more rooms than apartments, and single-family homes (especially large, high-priced homes) tend to be located in low-density suburbs.
>
> > "would benefit from some accommodation of sampling variation" and "Add some sort of resampling, e.g. bootstrap, to 3.2 to assess the sampling variation."
>
> We have added bootstrap-based error bars in the updated version, and observed that these intervals are extremely small relative to the analyzed correlations.
>
> > "The additional "tuning parameter" being the bandwidth in the kernel cdf may be beneficial" and "Investigate the usefulness of tuning the bandwidth using, e.g., cross-validation"
>
> Thank you for this suggestion. We have added this investigation to all experiments in Section 3.1, finding that cross-validation is able to effectively optimize the bandwidth.
>
> > "Use a log scale"
>
> Done.
>
> > "DBSCAN is supposed to "find noise" and so saying that identifying very small collections of points which should have been merged into clusters is rather unfair."
>
> For HDBSCAN, we actually excluded the samples it classified as noise, so actually our analysis gives it an unfair advantage, not an unfair disadvantage. We have clarified this in the updated version.

---

> > ### Comment · Reviewer_qhJo · 2023-09-26
> > **Thanks to authors**
> >
> > Thanks to the authors for providing the additional results and analysis in their revised manuscript. I actually don't currently have anything further.

---

### Review · Reviewer_2qmP · 2023-09-13

**Summary Of Contributions:**

The paper proposes a kernel density quantile transformation as a preprocessing step to be used as an alternative to min-max transformation or empirical quantile transformation. The paper briefly introduces the method and then documents on a set of synthetic and classical UCI experiments its properties.

**Audience:**

No

**Broader Impact Concerns:**

I do not have any concerns in this respect.

**Claims And Evidence:**

No

**Requested Changes:**

- Please address the concerns mentioned under Weaknesses.
- Equation (4) - are the bounds of the integral swapped? Please check and correct if necessary.
- Equation (4) - by $P(a, b)$ you mean $P(a \leq x \leq b)$? Please check and correct accordingly.
- On top of page 4 you suggest to estimate the variance $\sigma_X^2$ from the data and set just the multiple $\alpha$ as a hyper-parameter. Later on you say that $\alpha=1$ seems to perform the best for the badwidth. Why is this surprising. Isn't this to be expected (given that you estimate the variance)?
- Paragraph 2 page 3 - you claim the KD-quantile converges to min-max with $h \to \infty$ and to vanilla quantile with $h \to 0$. Please give the proofs.
- top of page 4 - Please explain what "vanishing bandwidth via Scott's Rule" actually is. I am sorry, but I do not know.
- section 3.1.1. - I am not clear on the classification pipeline (preprocess, pca, NB). Please elaborate, in particular clarify the purpose of the PCA step.
- section 3.1.1. - you claim that in Figure 2 we see the reduced distance of outliers - I may not be reading the figure correctly, but where do I see it? Please clarify.
- end of section 3.1.1 - you say KD is "non-inferior to the better of the two". Isn't it worth than min-max in (B) and than z-score in (C). Please check and correct if appropriate.
- Table 1 - the differences in the mean are negligible (compared to the std). Please comment / clarify.
- last phrase section 3.1.2 - we see ... I do not see. Please help me see.
- section 3.3 - Please clarify the motivation for the particular setup of the synthetic datasets. Are these supposed to illustrate something interesting about the method? Why have you chosen particularly these mixtures? Are these somehow particularly challenging for your or the other methods? How? Why?

**Strengths And Weaknesses:**

*Strengths*
The paper is well written, very easy to follow and grasp.

*Weaknesses*
The method seems somewhat ad hoc and it is not clear to me from the description and the experiments, what benefits it really brings.
For example paragraph 3 states - "we are not interested in estimating the true quantiles .. instead use the KDE to merely construct a preprocessing step". This is too vague a statement. Any transformation can be used as preprocessing step. Why is this supposed to be particularly good?
On the other hand, kernel methods are well known to outperform classical linear methods when the relationships cannot be expected to be simply linear. I am not clear from reading the paper, to what extent is the addition of the quantile tranformation (on top the kernel transform) really beneficial.
The argumentation in favor of the method is largely based on the results of experiments. These are, however, not very convincing to me. This may in part be by not stressing enough where does the strength of the models really come from - mainly from the kernel density estimation or from the qunatile tranform.

---

> ### Author Response · Authors · 2023-09-18
> **Re: Review of Paper1458 by Reviewer 2qmP**
>
> We thank you for your time and attention.
>
> > "Why is this supposed to be particularly good? On the other hand, kernel methods are well known to outperform classical linear methods when the relationships cannot be expected to be simply linear. I am not clear from reading the paper, to what extent is the addition of the quantile tranformation (on top the kernel transform) really beneficial."
>
> Just to avoid any possibility of confusion, we would like to clarify that we are using the kernel density estimator, and neither the kernel trick nor the quantile function. So there are not two transforms, a "kernel transform" and a "quantile transform". The KDE corresponds to a distribution, and our proposed approach involves evaluations of the cdf of this distribution. We have renamed the paper from "The Kernel Density Quantile Transformation" to "The Kernel Density Integral Transformation", so as to avoid giving the impression that we are using the quantile function (ie the inverse cdf).
>
> > "Equation (4) - are the bounds of the integral swapped"
>
> Thank you for pointing this out. It has been corrected in the updated version.
>
> > "Equation (4) - by P(a, b) you mean P(a <= x <= b)?"
>
> Yes, but we use the former notation to emphasize that we are treating this as a function of a and b.
>
> > "Isn't this to be expected (given that you estimate the variance)?"
>
> The optimal KDE bandwidth in terms of asymptotic mean integrated squared error (AMISE) -- for estimating the true density -- approaches zero as the sample size goes to infinity, with Scott's and Silverman's rule using h = o(N^{-0.2}). Our empirical results therefore indicate that improved downstream performance coming from our approach does not come from low-error estimation of the density, because estimators with suboptimal error can lead to improved downstream performance. We admit to finding this rather mysterious, and do not yet have a theoretical explanation of this effect.
>
> > "Paragraph 2 page 3 - you claim the KD-quantile converges.... Please give the proofs."
>
> We have added proofs to the Appendix of the revised manuscript (in Supplementary Material).
>
> > "Please explain what 'vanishing bandwidth via Scott's Rule' actually is."
>
> Added. It is h = N^{-0.2}\sigma_X. The N^{-0.2} factor comes from minimizing the AMISE, and the \sigma_X factor makes it invariant to rescaling of the data.
>
> > "I am not clear on the classification pipeline (preprocess, pca, NB). Please elaborate, in particular clarify the purpose of the PCA step."
>
> The full description and code is given in the reference (Raschka, 2014). We merely replicated that setup, rather than employing a perhaps-better pipeline, so as to avoid the possibility that we tweaked our setup to give our proposed approach an unfair advantage. Raschka (2014) observes that PCA substantially increased accuracy from 65%.
>
> >  "section 3.1.1"
>
> We have rewritten this section in the updated version.
>
> > "Table 1 - the differences in the mean are negligible (compared to the std). Please comment / clarify."
>
> On this dataset-of-datasets, inter-dataset variation is larger than inter-method variation. However, we added the results of paired Wilcoxon tests, showing that our improvements are statistically significant. Furthermore, our plot of relative ROC AUC vs sample size indicates substantial improvements on some datasets.
>
> > "last phrase section 3.1.2 - we see ... I do not see. Please help me see."
>
> We switched to a log-scaled x-axis in the updated version, which makes this more evident.
>
> > "section 3.3 - Please clarify the motivation for the particular setup of the synthetic datasets."
>
> Unfortunately, we are not aware of standard synthetic datasets for univariate clustering, so the selection of these was, to be honest, a bit arbitrary. While we selected datasets where previous methods struggled, we were unable to generate datasets where previous methods outperformed our approach.

---

> > ### Comment · Reviewer_2qmP · 2023-09-25
> > **More comments/questions/suggestions to the revised version - part I**
> >
> > Dear authors, thanks you for your responses and the updated version of the paper. I have carefully reviewed both and unfortunately, these trigger even more questions and comments.
> >
> > First and foremost, it is not clear to me what is actually the message of your paper. Is it simply saying that the KD integral transform is another possible pre-processing transformation one might consider? Or is it that simple Gaussian-KDE with bandwidth $h=1 \cdot \hat{\sigma}$ is the rule of thumb to go for? Or is it the efficient computation of the quantiles you introduce in section 2.3? Or is the clustering method you introduce in 2.5? I presume all of these to some extent. I recommend that these messages are made much clearer so that the reader can understand the "purpose" of the paper, rather than reading it as a collection of ad-hoc solutions.
> >
> > Related to this, it seems to me that the focus of the paper has changed quit a bit from the first writing. You have now introduced the poly kernel in section 2.3 and seem to be using it in most of the experiments (at least that seems to be the claim in the introduction of section 3). This seems as a rather significant change to me as the shape and the function form of the kernel is now different. This introduces an additional complexity for the user not only in choosing the actual kernel function but also, for example, the order of the polynomial. You seem to be picking K=4 as the way to go for your experiments without any discussion or justification. Yes, it may perform well on your experiments, but is this a result of out-of-the-box (would be surprising) or you searching over the K space or a reasonable complexity compromise? This discussion is missing.
> >
> > To your previous answers:
> > - What does asymptotically vanishing bandwidth with $N \to \infty$ have to do with the real small sample set up that you seem to be investigating. I find it not at all surprising that a larger bandwidth works fine in this setting. That is also the reason for all the rules of thumps that have been proposed for the bandwidth estimation. Perhaps I am missing something. If so, please explain this better in the paper. Also, you say in the end of the 3rd para in section 1 that you "choose h large ... so that ... deviate substantially from the estimated quantiles". However, you never explain what are these "estimted quantiles" and provide no comparisons for the reader to be able to judge, how much you really deviate.
> > - I appreciate you honesty in saying that you find the good performance to be "rather mysterious". I also do understand that scientific progress comes in steps and that sometimes the advancements cannot be all fully explained in one go. However, given that the simple bandwidth fix seems to be one of the contributions of your paper, I believe either some theory needs to be provided or the experimental evidence needs far more detail (see below some comments/suggestion for the experiments).

---

> > > ### Comment · Reviewer_2qmP · 2023-09-25
> > > **More comments/questions/suggestions to the revised version - part II**
> > >
> > > New comments/questoins/suggestions:
> > > - in the abstract you say you offer more "robustness to the weaknesses of each". What do you mean? I didn't find in the text any robustness discussion.
> > > - last sentence of section 2.2 "As we will see ... as the sample size $N \to \infty$". This is a very strong claim given that most of your experiments focus on small size datasets. Specifically designed experiments for showing this behaviour for really big N would corroborate your claims more convincingly.
> > > - Figure 1 - I do not understand the plots. On the x axis you indicate the time in sec. How come the time changes? Is the time a surrogate for the number of reference points? Isn't the number of reference points fixed as $R = min(R_{max} = 1000, N)$? Why is the poly kernel represented by just a single point? It is not treated similarly as the Gaussian kernel?
> > > - Is the conclusion of section 2.3 that you actually recommend using the poly kernel? This is quite a turn from your first draft of the paper.
> > > - As in one of the earlier comments from one of the reviewers, the bandwidth seems to be a critical hyperparameter of your methods. Nevertheless, you do not show it in the poly kernel formulation. Please give the full form of the poly kernel including the bandwidth. Also, what about the order of the polynomial K? I believe the choice of K deserves at least some discussions.
> > > - Section 3.1.1 - what is the criteria for the bandwidth selection in the CV approach? I'm guessing it is likelihood-based (otherwise why would not it choose the best bandwidth)?
> > > - Section 3.1.2 - thanks for adding a regression example. The z-transform seems to be missing in Figure 4, any reason? Further "the magnitude of improvement ... was larger". To me it makes no sense to compare the magnitudes of errors in a regression problem with accuracy in a classification problem. This claim seems vacuous.
> > > - Figure 3 - the green curve for KD-integral has bars at every point. My first interpretation of these was that these are quartiles from your 100 train-test splits. But it seems, they have no actual meaning (just a visual thing). Right? Very deceiving and confusing.
> > > - Section 3.1.3 - you say that these are 142 datasets, all with 50 samples only. How can you in Figure 5 plot the performance for $10^2 - 10^5$ samples? I'm confused.
> > >
> > > In conclusions, as much as I appreciate that your method seems to work rather well on a number of experiments, I feel that the paper would benefit greatly from more rigor, more informative discussion of the critical choices for the kernel and bandwidth selection, and preferably some theory-based justification or at least more support from carefully "designed" experiments to demonstrate the performance in controlled conditions.

---

> > > > ### Author Response · Authors · 2023-09-27
> > > > **Re more comments - part II**
> > > >
> > > > > in the abstract you say you offer more "robustness to the weaknesses of each". What do you mean? I didn't find in the text any robustness discussion.
> > > >
> > > > We have replaced "robustness to" with "protection from" to clarify that we do not mean robustness in the statistical sense (eg "robust estimator"), but rather that with $\alpha=1$, we typically observe performance similar to (or better than) the better of the two. Thanks for pointing out this potentially confusing terminology.
> > > >
> > > > > Specifically designed experiments for showing this behaviour for really big N would corroborate your claims more convincingly.
> > > >
> > > > We have added experiments to V3 (Section 3.1.2 and Appendix A.3) utilizing the CA Housing (N=20640), where we randomly subsample to (1000, 2000, 5000, and 10000) in each simulation. The optimal bandwidth does not depend on sample size, at least for this dataset.
> > > >
> > > > > Figure 3 - the green curve for KD-integral has bars at every point. My first interpretation of these was that these are quartiles from your 100 train-test splits.
> > > >
> > > > The error bars depict +/- standard deviation over train-test splits; we now note this in the figure caption.
> > > >
> > > > > Is the conclusion of section 2.3 that you actually recommend using the poly kernel? This is quite a turn from your first draft of the paper.
> > > >
> > > > Yes, and we have clarified this in V3 "Practical Recommendations". This makes sense, since the poly-exp order and bandwidth are chosen to make it closely approximate to the Gaussian kernel.
> > > >
> > > > > Please give the full form of the poly kernel including the bandwidth. Also, what about the order of the polynomial K?
> > > >
> > > > We have added more thorough discussions in Section 2.3 and A.2 in V3.
> > > >
> > > > > Section 3.1.1 - what is the criteria for the bandwidth selection in the CV approach? I'm guessing it is likelihood-based (otherwise why would not it choose the best bandwidth)?
> > > >
> > > > It is actually accuracy. But the best bandwidth does vary over simulated train-test splits, especially for small-N (eg Wine & Iris). And, because we choose bandwidth independently in each simulation via an inner CV split, rather than peeking at other simulations or at the test data, the picked bandwidth also varies. These two forms of variation combine so that the CV approach shown in pink does not always attain the peak test accuracy shown in green.
> > > >
> > > > > Section 3.1.3 - you say that these are 142 datasets, all with 50 samples only.
> > > >
> > > > Sorry, this was a typo. We have replaced "each with at 50 samples" with "each with at least 50 samples" in V3.
> > > >
> > > > > The z-transform seems to be missing in Figure 4, any reason?
> > > >
> > > > It's actually there, but overlaps with min-max scaling, hence the toothed curve. This is expected for linear regression with an intercept but without regularization, since both transformations are linear.
> > > >
> > > > > Further "the magnitude of improvement ... was larger". To me it makes no sense to compare the magnitudes of errors in a regression problem with accuracy in a classification problem.
> > > >
> > > > We have replaced the above claim with the following: "In both regression datasets, the performance improvement o ffered by KD-integral exceeds the di fference in performance between linear and quantile transformations."

---

> > > ### Author Response · Authors · 2023-09-27
> > > **Re more comments - part I**
> > >
> > > We appreciate your careful feedback on V2 of the manuscript, and have incorporated it into V3.
> > >
> > > > First and foremost, it is not clear to me what is actually the message of your paper.
> > >
> > > We have added a "Practical Recommendations" section to the Discussion to address this:
> > >
> > > "We recommend that if one must employ a feature preprocessor without any tuning or comparisons between different preprocessing methods, then KD-integral with bandwidth-factor of 1 is the best one-shot option. If instead tuning is possible, we suggest comparing the performance of KD-integral, with a log-space sweep of α ∈ [0.1, 10], in addition to the other popular preprocessing methods. Finally, we recommend that one always estimate our transformation with the order-4 polynomial-exponential kernel approximation of the Gaussian kernel, and represent it with Rmax = 1000 reference points."
> > >
> > > Relevant to the above recommendation, in V2 we added the results of Wilcoxon signed-rank tests to Table 1, showing statistically significant improvements on the Small Dataset Benchmark.
> > >
> > > > You have now introduced the poly kernel in section 2.3
> > >
> > > It is shown in (Hofmeyr, 2019 -- Figure 3) that the poly-exponential kernel with K=4 (and a rescaling factor to convert from a given Gaussian kernel bandwidth to a poly-exp kernel bandwidth) is an exceedingly good approximation of the Gaussian kernel. Hofmeyr (2019) notes that the integrated squared distance between the two kernels is \approx 3e-4, and our analysis in Figure 1 lends further support to the closeness of the approximation. In addition, we reran all the other experiments in the paper with K=4, and observed no differences, either visually or in the reported quantitative metrics.
> > >
> > > We have added an explanation of this to the main text, and details on bandwidth rescaling to the Appendix A.2 of V3.
> > >
> > > > What does asymptotically vanishing bandwidth with $N \rightarrow \infty$ have to do with the real small sample set up that you seem to be investigating
> > >
> > > The non-vanishing bandwidth empirically succeeds on both small-N and large-N settings, both of which we are investigating in our experiments. For example, for California housing (N=20640) in Figure 4A, the optimal bandwidth factor exceeds 1. (Also, please see our next comment regarding additional experiments.) And for univariate clustering, we fixed the bandwidth factor at 1 while varying N from 100 to 5000.
> > >
> > > > However, you never explain what are these "estimated quantiles" and provide no comparisons for the reader to be able to judge, how much you really deviate.
> > >
> > > We have updated the paper to say "empirical quantiles" rather than "estimated quantiles". These are compared on the y-axes in Figure 2 B vs C, and also on the second vs third rows of Figures 8 and 9.  We have added pointers in the main text to highlight these to readers.

---

### Review · Reviewer_2B75 · 2023-09-16

**Summary Of Contributions:**

This paper proposes to use the Gaussian kernel density estimation for feature preprocessing. Each feature value is transformed based on the definite integral over the KDE. The proposal has been empirically evaluated on several scenarios including classification and univariate clustering on synthetic and real-world datasets.

**Audience:**

Yes

**Broader Impact Concerns:**

I do not have any concerns.

**Claims And Evidence:**

No

**Requested Changes:**

Please resolve all the above weaknesses.

**Strengths And Weaknesses:**

### Strengths

- Feature engineering is an important issue in machine learning, thus the objective of this paper is relevant.

### Weaknesses

- The novelty of this paper is limited. The KDE with the Gaussian kernel is a quite standard method, and using it for feature preprocessing is also a simple application. Of course, it is acceptable if such a simple approach leads to some unexpected theoretical properties and/or remarkable empirical advantages. However, this paper is not the case.
- Motivation is not clear. What is the current challenge to be addressed, and for which application is the proposal required?
- Presentation should be improved. For example, although the kernel bandwidth $h$ is one of the most important parameters in the proposal, it is missing in Equation (3).
- In experiments, accuracy without any feature preprocessing should be compared as a baseline.
- In experiments, in Table 1, the proposal is simply not significantly better than other methods. By considering that the proposal requires an additional parameter $h$ (or $\alpha$), its practical advantage is not convincing.
- In experiments, clustering on univariate data does not add much information as it is a too simple problem. Trying clustering on multivariate data might be interesting.

---

> ### Author Response · Authors · 2023-09-18
> **Re: Review of Paper1458 by Reviewer 2B75**
>
> We thank you for your time and attention.
>
> > The novelty of this paper is limited. The KDE with the Gaussian kernel is a quite standard method, and using it for feature preprocessing is also a simple application.
>
> With the caveat that "interestingness" and not "novelty" is the criteria for TMLR, we note that while using the KDE for feature preprocessing is conceptually simple and obviously possible in hindsight, it was not (to our knowledge) previously proposed or analyzed. Shifting the focus to interestingness, we argue that our paper will be of primary interest to practitioners, since they can shift to our method (with $\alpha=1$) with no additional tuning cost, for a substantial chance of improved performance and little chance of reduced performance.
>
> > "Motivation is not clear. What is the current challenge to be addressed, and for which application is the proposal required?"
>
> We have revised the abstract and introduction in the updated version.
>
> > "For example, although the kernel bandwidth is one of the most important parameters in the proposal, it is missing in Equation (3).
>
> Thank you for your recommendation; this has been revised in the updated manuscript.
>
> > "In experiments, in Table 1, the proposal is simply not significantly better than other methods. By considering that the proposal requires an additional parameter, its practical advantage is not convincing."
>
> Our results in Table 1 (noted both in Section 3 and elsewhere) did not involve tuning an additional hyperparameter at all. That the default setting frequently performs substantially better, and rarely performs substantially worse, suggests that in applied settings one may adopt the default ($\alpha=1$) as a robust alternative to trying various preprocessing approaches, and to tuning the hyperparameter of our approach. (In our updated revision, we added analysis of our approach with cross-validation, at the request of Reviewer qhJo. Nevertheless, the main observation of Table 1 is that KDI with $\alpha=1$ makes sense as a default no-downside choice for practitioners.)
>
> > "In experiments, clustering on univariate data does not add much information as it is a too simple problem."
>
> As our experiments show, previous methods do in fact struggle on this allegedly "simple problem", so we disagree with the premise of this statement.

---

> > ### Comment · Reviewer_2B75 · 2023-09-26
> >
> > Thank you for your reply. The manuscript improves and my concerns are addressed.

---

### Comment · Action_Editors · 2023-09-19
**Please respond or reflect on author comments**

Dear reviewers.  Thank you all of your reviews.

You all had some concerns, please take a look at the author's comments and make at least a quick note on if it sufficiently addressed your concerns enough to make this paper meet the bar for TMLR.  That is, will it be interesting for the TMLR audience (2/3 said yes) and do they support their claims with evidence (0/3 said yes).

thanks!

---

### Decision · Action_Editors · 2023-10-06

**Recommendation:** Accept as is

**Comment:**

The decision among the reviewers was mixed.  But I feel that the paper has merit, and does a sufficient job to provide evidence that the technique can be useful.  Such pre-processing is common enough, and there is not always a good solution that I think the proposed approach is a useful to consider.

**Audience:**

The reviewers were again mixed on this.  However, I think this is a simple idea that can sometimes be the appropriate approach for this problem, and scaling such as this is a common way to deal with unknown ways to compare variables.  Hence, I do believe other ML researchers will find this approach useful.

**Claims And Evidence:**

The reviewers were somewhat mixed on this point.  However, I recognize that this sort of heuristic is hard to completely verify as always being the best.  My opinion are the experiments are enough to show this method has merit, and will sometimes, if not often be the right tool for this sort of pre-processing.